# In Silico Identification of Peptides with PPARγ Antagonism in Protein Hydrolysate from Rice (*Oryza sativa*)

**DOI:** 10.3390/ph16030440

**Published:** 2023-03-15

**Authors:** Felipe de Jesús Ruiz-López, Bryan Alejandro Espinosa-Rodríguez, David Arturo Silva-Mares, Blanca Edelia González-Martínez, Manuel López-Cabanillas Lomelí, Luis Fernando Méndez-López, Jesús Alberto Vázquez-Rodríguez

**Affiliations:** 1Universidad Autónoma de Nuevo León, Facultad de Salud Pública y Nutrición, Centro de Investigación en Nutrición y Salud Pública, Monterrey 66460, Mexico; 2Universidad Autónoma de Nuevo León, Facultad de Ciencias Químicas, Laboratorio de Farmacología Molecular y Modelos Biológicos, Monterrey 64570, Mexico; 3Universidad Autónoma de Nuevo León, Facultad de Medicina, Departamento de Química Analítica, Monterrey 64460, Mexico

**Keywords:** protein hydrolysates, bioactive peptides, rice protein, docking, PPARγ

## Abstract

At least half the population in industrialized countries suffers from obesity due to excessive accumulation of adipose tissue. Recently, rice (*Oryza sativa*) proteins have been considered valuable sources of bioactive peptides with antiadipogenic potential. In this study, the digestibility and bioaccessibility in vitro of a novel protein concentrate (NPC) from rice were determined through INFOGEST protocols. Furthermore, the presence of prolamin and glutelin was evaluated via SDS-PAGE, and their potential digestibility and the bioactivity of ligands against peroxisome proliferator-activated receptor gamma (PPARγ) were explored by BIOPEP UWM and HPEPDOCK. For the top candidates, molecular simulations were conducted using Autodock Vina to evaluate their binding affinity against the antiadipogenic region of PPARγ and their pharmacokinetics and drug-likeness using SwissADME. Simulating gastrointestinal digestion showed a recovery of 43.07% and 35.92% bioaccessibility. The protein banding patterns showed the presence of prolamin (57 kDa) and glutelin (12 kDa) as the predominant proteins in the NPC. The in silico hydrolysis predicts the presence of three and two peptide ligands in glutelin and prolamin fraction, respectively, with high affinity for PPARγ (≤160). Finally, the docking studies suggest that the prolamin-derived peptides QSPVF and QPY (−6.38 & −5.61 kcal/mol, respectively) have expected affinity and pharmacokinetic properties to act as potential PPARγ antagonists. Hence, according to our results, bioactive peptides resulting from NPC rice consumption might have an antiadipogenic effect via PPARγ interactions, but further experimentation and validation in suitable biological model systems are necessary to gain more insight and to provide evidence to support our in silico findings.

## 1. Introduction

Obesity is a medical condition characterized by an excessive accumulation of fat in adipose tissues arising from excess calorie intake due to a discrepancy between energy intake and energy usage [1]. Excess adiposity is associated with the development of metabolic diseases related to chronic inflammation, including type 2 diabetes, hypertension, cancer, and cardiovascular disorders [2]. In principle, obesity is multifactorial, involving genetic, psychological, economic, and social factors, which can alter the adipocyte physiology, thus causing the synthesis and release of different compounds that can alter other pathways [3], generating hypertrophy (cell number increase) or hyperplasia (cell size increase). One of the most studied mechanisms related to adipogenesis is peroxisome proliferation factors (PPARs), especially PPAR-γ. PPAR-γ is responsible for regulating the gene expression of enzymes involved in the storage of fatty acids in adipose tissue, such as acyl-CoA synthase, lipoprotein lipase, and phosphoenolpyruvate carboxykinase [4]. Due to this, PPAR-γ is principally involved in lipogenesis, adipocyte differentiation, cell proliferation, and insulin sensitivity [5], among other metabolic processes. Hence, adipogenesis is mainly controlled by the peroxisome proliferator-activated receptor γ, which upon activation promotes preadipocyte differentiation into mature adipocytes [6]. Therefore, obesity may be controlled by reducing adipogenesis through inhibition of PPARγ activity to negatively modulate preadipocyte differentiation [4]. Studies with PPARγ-gen knock-down animal models result in a protective effect on the development of diseases linked to overweight and obesity [7]. Furthermore, drugs with PPARγ agonism that stimulate adipogenesis are employed for the pharmacological management of hyperglycemia in type 2 diabetes [8]. However, despite their clinical effectiveness, long-term consumption is frequently accompanied by several unwanted side effects (weight gain, edema, and congestive heart failure) that may exceed the benefits [9].

Therefore, in recent years, there has been an increased interest in the identification of bioactive peptides derived from edible plants with potential functionality against obesity and minimal adverse reactions [10,11,12]. Food proteins and their bioactive peptides (BAP), a product of a monogastric digestive process, present biological activities against the inhibition of the angiotensin-converting enzyme (ACE) (EC 3.4.15.1), a central component of the renin–angiotensin system, which regulates blood pressure; inhibition of dipeptidyl peptidase IV (DPPIV) (EC 3.4.14.5), an enzyme involved in glucose homeostasis, and is the therapeutic target in type 2 diabetes; and presents antioxidative activity [13] and stimulates lipolysis and increased insulin sensitivity, while diminishing inflammation [14,15], and some BAP had also been found to be involved in PPARγ inhibition or regulation. The antiadipogenic effect via PPARγ is possible due to the interaction of this receptor in adipose tissue, and, additionally, it has been reported that peptides that exert bioactivity are usually hydrophobic in nature [16]. These PPARγ BAPs are mainly studied in whey protein, spirulina (*Spirulina platensis*), and legumes such as soy (*Glycine max*) and tepary bean (*Phaseolus acutifolius*) [15,17,18], but cereal proteins such as rice (*Oryza sativa*), mainly prolamin and glutelin fractions, present activity against ACE and DPPIV enzymes, and present antioxidant activity [19,20,21], and their potential bioactivity against specific receptors related to obesity, such as PPARγ, remains unknown.

In this regard, bioinformatic studies, such as docking techniques, present an economical, reliable, and straightforward approach to identify candidate ligands against key proteins involved in the process of adipogenesis. Databases of BAPs are reported, providing information necessary to construct new algorithms for predicting biological functions [22,23]. This article provides molecular evidence for the potential ligand bioactivity of peptides derived from the in silico hydrolysis of rice proteins, specifically glutelin and prolamin fractions, that showed high affinity for the antiadipogenic areas of the PPARγ receptor.

## 2. Results

### 2.1. Proximate Composition and In Vitro Digestibility

The novel protein concentrate (NPC) of rice shows an amino acid profile with elevated values of glutamine (18.39 g/100 g protein), sulfur amino acids (Cys-Met 8.46 g/100 g protein), and important values of histidine, leucine, proline, aromatic amino acids (PHE + TYR), and tryptophane (Table 1). The use of an in vitro model simulating gastrointestinal digestion showed a 43.07% protein digestibility (IVPD) and mimicking of an epithelial barrier in the dialysis phase, with an estimated 35.92% of protein bioaccessibility (IVBA).

### 2.2. Electrophoresis (SDS-PAGE)

Figure 1 shows the NPC separated by molecular weight in the presence of SDS. We could observe that the predominant proteins are those with molecular weights between 50–60 kDa (glutelin fraction) and 10–20 kDa (prolamin fraction).

### 2.3. In Silico Digestion

The proteolytic process with pepsin, trypsin, and chymotrypsin released 85 and 42 peptides from glutelin and prolamin, respectively. According to their greater absorption, peptides that presented a length between two and five amino acids were evaluated for their affinity towards PPARγ with HPEPDOCK (Table 2) [25,26].

### 2.4. In Silico Screening for Bioactive Peptides against PPAR-γ

High affinity for PPAR-γ was considered when the calculated scoring bonding results ≤160 kcal/mol according to Yao et al., 2015 [27]. Hence, the preliminary screening using the HPEPDOCK server suggests that two and three peptides from glutelin and prolamin, respectively, are candidates to bind with high affinity to the receptor PPARγ (Table 3). Therefore, docking studies in Autodock Vina were performed to evaluate their binding affinity to PPARγ regions associated with antiadipogenic effects. Molecular coupling analysis showed that peptides IVPQH and PIVF derived from glutelin presented binding energies of −1.87 and −2.18 kcal/mol, respectively, with molecular interactions to Cys285 for IVPQH, and Ser289 and Cys285 for PIVF. For prolamin, the peptides QPY and QSPVF presented a binding energy of −5.61 and −6.38 kcal/mol with molecular interactions to Cys285, Ser289, and Tyr473 for QPY; and Cys285 and Gly284 for QSPVF. The control molecule (GW9662) has been widely reported to have irreversible antagonistic action in PPARγ and showed the highest binding energy (−7.98 kcal/mol). GW9662 covalently joins the amino acid residue Cys285, which together with Phe264, Hist266, Ile281, and Met348, are the most frequent regions that bind the antiadipogenic molecules [28] (Table 3).

Figure 2 shows the interaction points towards the PPARγ region associated with antiadipogenic effects. IVPQH, PIVF, and IIQGR peptides of the glutelin fraction present hydrogen bond formation. For the IVPQH pentapeptide, the interaction went towards the amino acid residue Cys285, for the tetrapeptide PIVF, it was in Cys285 and Ser289; moreover, for IIQGR, it was in Arg288.

The peptides released from the prolamin fraction showed bond formation for the QPY tripeptide toward Cys285, Ser289, and Tyr473. For the QSPVF pentapeptide, link formation was toward amino acid residues Cys285 and Gly284.

### 2.5. Pharmacokinetics and Drug-Likeness Prediction

Table 4 shows the values obtained for each of the parameters considered in the Lipinski rule. The molecular weight to be presented by an absorbable molecule according to the Lipinski rule is ≤500 g/mol, in which, glutelin PIVF peptides and QPY peptides from prolamin were those which obtained molecular weights that met this parameter. In the assessment of lipophilicity by means of n-Octanol/Water Partition Coefficient (iLogp), all molecules obtained values ≤ 5. In numbers of H-bond donors (HBDs), only QPY tripeptide from prolamin obtained a value of ≤10.6 and, finally, all ligands obtained values of numbers of H-bond acceptors ≤ 10.6. According to the Lipinski rule, a molecule is considered active/absorbable orally if it does not violate two or more parameters of the rule. Hence PIVF and QPY peptides from glutelin and prolamin fractions, respectively, are considered active/absorbable orally.

## 3. Discussion

The amino acid profile of NPC of rice is consistent with the literature, obtaining higher values in essential amino acids with respect to FAO reference, except Lysine (LYS). Moreover, NPC contains high proportions of non-polar amino acids such as histidine, isoleucine, leucine, and proline, associated with different bioactivities [29]. Figure 1 indicated the different protein fractions of NPC obtained from rice; glutelin between 50–60 kDa and prolamin fraction between 10–20 kDa were observed. These data are consistent with references, in which glutelin is observed to be 57 kDa of molecular weight, and prolamin between 12 to 17 kDa [30,31].

Although animal proteins have a higher quality in terms of digestibility and essential amino acids, it is important to note that the bioaccessibility and bioavailability of plant proteins is increased by subjecting proteins to heat and exposure to acids or alkaline conditions, since these processes alter the molecular and supramolecular structures of proteins, which allows digestive enzymes to gain greater accessibility [32], helping to obtain peptides with two to five amino acids, which are transported passively [25,33]. Hence, the peptides studied in this research had a length between two and five amino acid residues and were measured by ligand–receptor affinity [25,26].

Peptides that presented a docking score on the HPEPDOCK server below −160 kcal/mol were considered as having high affinity against the PPARγ receptor. The affinity of the ligand GW9662 was not calculated by the HPEPDOCK software since it was designed to measure blind protein–peptide docking [34]. The reports mentioned that the determination of binding energy using HPEPDOCK server software exhibits a good correlation with results obtained experimentally [35,36]. However, it is necessary to know the energies and regions to which the ligands join the receptor PPARγ.

The results of docking using Autodock Vina showed that all candidate ligands form intermolecular links to sites with adipogenesis antagonistic effects (Cys285, Arg288, Ser289) [28,37]. Nevertheless, it was necessary to consider the binding energies, which are between −1.87 and −6.38, with the highest affinity presented for QSPVF and QPY peptides, both prolamin fractions, with binding energies of −5.61 and −6.38 kcal/mol, respectively. Similarly, the results obtained by the molecule GW9662, the antiadipogenic effect via PPARγ of which has been widely reported in the scientific literature, show similar values of binding energy (−7.98 kcal/mol) and with the interaction of hydrogen bonds at antiadipogenic sites (Ser289 and Cys285). Thus, referencing the energy and interaction sites of this GW9662, it can be established that the compounds QPY and QSPVF could have antiadipogenic activity [38,39].

All the peptides analyzed present amino acids with hydrophobic lateral chains. In silico, in vitro, and in vivo studies have reported that molecules that present nonpolar and hydrophobic amino acids (VAL, LYS, ILE, ALA, TRP, MET, PRO, and, especially, PHE) are crucial for their functionality as bioactive compounds for hydrophobic interactions [14,40,41].

Numerous studies are conducted at the in vitro level with the cell line 3T3-L1, indicating that plant peptides can exert antiadipogenic bioactivity. For example, the RLLPH pentapeptide from hazelnut (*Corylus hazelnut*) showed an 82.06% decrease in lipid accumulation compared to cells that were not treated [42]. FFL, LLSL, QQEG, and LVLL peptides, which have hydrophobic amino acids in both terminations, except for the QQEG tetrapeptide, have a lipid accumulation inhibition of 13–28% in studies with the 3T3-L1 cell line and, also, hydrophobic regions affect the reduction of blood triglycerides, as well as lipid stimulation in fat cells [14]. Similarly, peptides from quinoa (*Chenopodium quinoa*) exhibit antiadipogenic activity proportional to the number of peptides exposed to this cell line [43].

It is crucial to consider the bioavailability of the molecules studied when they are ingested. For example, two peptides could be absorbable when consumed orally: PIVF for glutelin and QPY for prolamin. For PIVF molecular weight, Ilogp and HBAs were the parameters that met this guideline and, for the QPY tripeptide and the control, all criteria were met. However, in the small intestine, other transports are responsible for absorber peptides, such as PepT1 and PepT2 (SLC15A1) for dipeptides and tripeptides [44]. In addition, another protein transporter called SOPT1/SOPT2 is responsible for transporting peptides with a length of four to five amino acid residues [45].

## 4. Materials and Methods

### 4.1. Novel Protein Concentrate (NPC) Obtention

Broken rice was mixed with 0.1% NaOH solution, stirred at room temperature for 1 h, and then left overnight. The mixture was centrifuged at 8000× *g* for 10 min. The supernatant was adjusted to pH 4.8 to precipitate rice protein and centrifuged at 8000× *g* for 20 min. Afterward, rice protein prepared by alkaline extraction was recovered by dispersing the protein precipitate in distilled water (1:10, *w*/*v*), neutralizing to pH 7.0, and freeze-drying [46].

### 4.2. Amino Acid Score

The determination of the amino acid score of the samples was through the method by [47]. Tryptophan was identified using the method of Dávila and Martínez with Betancur-Ancona’s modifications [48].

### 4.3. Electrophoresis (SDS-PAGE)

The protein fractions of NPC from rice were analyzed by denaturing electrophoresis in polyacrylamide gels (SDS- PAGE). Following the method proposed by Laemmli with some modifications [49]: 10 mg of each sample was dissolved in 1 mL of sodium dodecyl sulfate solution (SDS, 1% *w*/*v*). Five microliters of BenchMark Protein Ladder^TM^ was the molecular weight marker. The proteins were separated (Bio-Rad Mini-PROTEAN model 1658000EDM, Hercules, CA, USA) by a vertical electrophoresis chamber in Mini Geles Teo-Tricine SDS 4–12% RunBlueTM for 2 h at 80 V (1 h) and 100 V (1 h). The separated proteins were stained with staining solution Coomassie R-250 blue (0.1%), 40% methanol, and 10% glacial acetic acid, and gently stirred for 2.5 h, then faded with a methanol solution (40%). The images were photo-documented and analyzed.

### 4.4. In Vitro Digestibility and Bioaccesibility

Simulation of human gastrointestinal digestion (GIS) was conducted in vitro according to INFOGEST [50]. The entire GIS digestion steps were performed sequentially in 3 phases, at 37 °C. The pH was adjusted with HCl or NaOH (4 M) during digestion. We took aliquots before digestion (0 min); after 1, 5, 30, and 60 min of gastric phase; and after 1, 5, 30, and 120 min of duodenal phase. After each digestion time, the digested samples were filled with deionized water to 14 mL and then heated in boiling water for 5 min to stop the enzymatic digestions. The digested samples were centrifuged (10,000× *g*, 20 min, 4 °C) except those prepared for particle size distribution. The supernatants were kept frozen at −20 °C until use to determine the protein digestibility (%) for the Dumas methodology (920.06) [51] and calculated by:$$\mathrm{IVPD}\;\left(\%\right)=\frac{\%\mathrm{FP}}{\%\mathrm{IP}}\times 100$$
where IVPD (%): in vitro protein digestibility; FP: protein percentage at 120 min duodenal phase; IP: protein percentage at 0 min.

The absorption processes were simulated employing a static dialysis procedure with a cellulose membrane following the method described by Managa et al. (2021) [52] with slight modifications by Liu et al. (2021) [53]. First, the digested intestinal phase samples (8 mL) were poured into a tubular cellulose membrane for dialysis (D9652, Sigma-Aldrich, Darmstadt, Germany), previously hydrated in distilled water for 10 min, as a simplified model of the epithelial barrier. Then, each dialysis bag was placed inside a 125 mL flask and was totally immersed with 40 mL of SIF. This mixture was kept covered at 37 °C in a water bath for 120 min with gentle manual shaking every 15 min. The formula used to calculate the bioaccessibility of the digested protein was:$$\mathrm{IVPB}\;\left(\%\right)=\frac{\mathrm{Ptd}\;\left(\mathrm{mg}/\mathrm{mL}\right)}{\mathrm{Pti}\;\left(\mathrm{mg}/\mathrm{mL}\right)}\times 100$$
where IVPB (%) is the bioaccessible fraction in percentage; Ptd is the dialyzed protein of the digested samples; Pti is the protein content in the initial undigested sample.

### 4.5. In Silico Digestive Process

Protein fractions (prolamin (ID:1153), glutelin (ID:1536)) were obtained from the BIOPEP UWM protein database [54]. The process of in silico hydrolysis was performed using the digestive proteolytic enzymes pepsin (EC 3.4.23.1), trypsin (EC 3.4.21.4), and chymotrypsin (EC 3.4.21.1) [43] (Table 5). Peptide long-chains between 2 and 5 amino acids released during in silico digestion were analyzed [54,55].

### 4.6. Ligand Preparation

The GW9662 molecules were used as controls and were extracted from the PubChem database [56]. The peptides released from the digestive process with lengths between 2 and 5 amino acid residues were used for the subsequent tests. First, the ligand–receptor affinity was evaluated using the HPEPDOCK software, considering binding energy of less than −160 as high affinity. Subsequently, these high-affinity peptide ligands were converted to SMILES (Simplified molecular-input line-entry system) formats using the Dendrimer Builder tool. The ligands were imported into the Avogadro software 1.2.0 to a simulated molecular mechanism force field (MMFF94) under physiologic pH (pH 7.4) and subsequent identification of in silico antiadipogenic activity at the PPARγ was performed.

### 4.7. Receptor Preparation

The PPARγ receptor was extracted in its crystallized form from the PDB protein database (code: 3V9V). This protein binds to specific regions of DNA involved, mainly, in the process of adipocyte differentiation, in the form of a heterodimer with the retinoid X receptor (RXR) regulating the expression of genes involved in adipose tissue metabolism. This protein is the one that is most used for the identification of antiobesogenic compounds [10].

The affinity analysis of the ligands in the receptor was performed using the Autodock tools program, with an interaction area centered at 7745 × 50,606 × 57,552 and with dimensions of X = 70, Y = 40, and Z = 40, with a spacing of 0.0375 nm to cover those regions of the PPARγ receptor involved with antiadipogenic effects (Phe264, His266, Ile281, Cys285, Arg288, Ser289, Met348 e His449) (Figure 3) [28,37].

### 4.8. ADME Prediction

The ADME properties (absorption, distribution, metabolism, and excretion) analysis of molecules with antiadipogenic potential were performed using SwissADME^®^ software. The Lipinski Rule the “most well-known rule-based filter” was considered a parameter to know whether a compound is well absorbed orally or not. According to Lipinski’s rule, a molecule can be orally active/absorbable if it does not violate two or more of the following parameters [57,58]: molecular weight (MW) ≤500, octanol/water partition coefficient (iLOGP = A log P) ≤ 5, number of hydrogen bond donors (HBDs) ≤ 5, and number of hydrogen bond acceptors (HBAs) ≤ 10.6.

## 5. Conclusions

The NPC of rice presents values of amino score, digestibility, and bioaccessibility similar to those previously reported for rice and other protein concentrates from cereals. Furthermore, this research shows that prolamin and glutelin protein fractions in the NPC from rice are predicted to contain five bioactive peptides against PPARγ. In addition, the docking studies suggest that the prolamin-derived peptides QSPVF and QPY have affinity and pharmacokinetic properties to act as potential PPARγ antagonists. However, further experimentation and validation of the results are necessary to gain more insight and to provide evidence to support our in silico findings.

## Figures and Tables

**Figure 1 pharmaceuticals-16-00440-f001:**
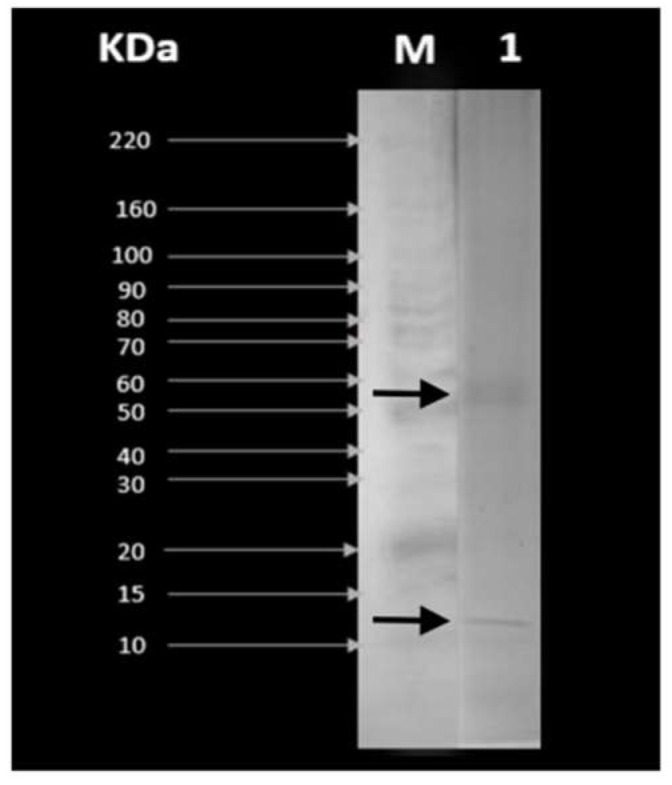
SDS-PAGE patterns of (1) novel protein concentrate NPC of rice (*Oryza sativa*). M is the molecular weight marker (control). First arrow (above to low) identifies glutelin fraction (57 kDa) and second arrow the prolamin fraction (12 kDa).

**Figure 2 pharmaceuticals-16-00440-f002:**
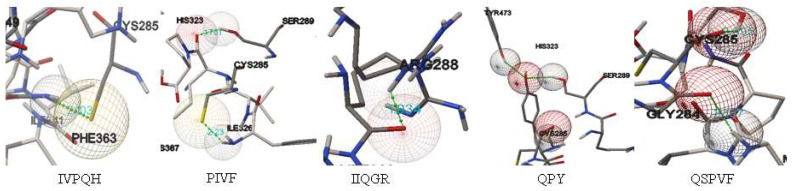
Intermolecular bonds of ligand–receptors.

**Figure 3 pharmaceuticals-16-00440-f003:**
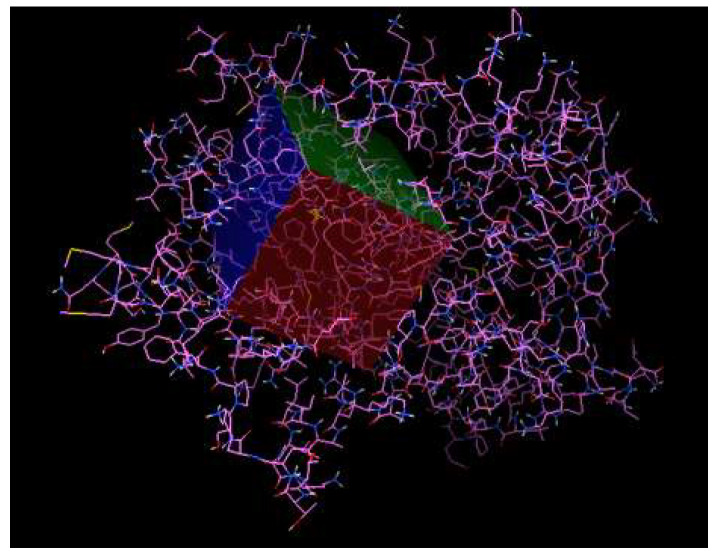
PPARγ’s active region site.

**Table 1 pharmaceuticals-16-00440-t001:** Amino acid composition of novel protein concentrate (NPC) of rice (*Oryza sativa*).

Amino Acid (g/100 g Protein)	NPC ^1^	FAO Ref +
ALA	5.75	--
ARG	8.36	--
ASP	8.05	--
GLN	18.39	--
GLY	4.70	--
PRO	4.19	--
SER	3.66	--
HIS	2.19	1.6
ILE	3.87	3.0
LEU	8.78	6.1
LYS	3.87	4.8
CYS + MET	8.46	2.3
PHE + TYR	5.22	4.1
THR	2.30	2.5
TRP	1.88	0.66
TOTAL PROTEIN (%)	83.40	--
IVPD (%)	43.07	--
IVBA (%)	35.92	--

^1^ Data presented as the mean, n = 3. + = [24].

**Table 2 pharmaceuticals-16-00440-t002:** In silico hydrolysis of the glutelin and prolamin proteins of rice (*Oryza sativa*).

Protein Fraction	Length of Peptides	Number of Peptides	%
Glutelin	1	66	77.65
	2 a 5	16	18.82
	>5	3	3.53
Prolamin	1	18	42.86
	2 a 5	14	33.33
	>5	10	23.81

**Table 3 pharmaceuticals-16-00440-t003:** Bonding energies and intermolecular bonds in antiadipogenic sites of PPARγ.

Fraction	Peptide	Length	Scoring Bonding HPEPDOCK	Bonding Energy (kcal/mol)	Intermolecular Bonds
Glutelin	IVPQH	5	−170.40	−1.87	CYS285
	PIVF	4	−188.22	−2.18	SER289 CYS285
	IIQGR	5	−170.12	−2.07	ARG288
Prolamin	QPY	3	−168.94	−5.61	CYS285 SER289 TYR473
	QSPVF	5	−175.73	−6.38	CYS285 GLY284
Control	GW9662	---	---	−7.98	CYS285

Data presented as the mean, n = 100.

**Table 4 pharmaceuticals-16-00440-t004:** ADME prediction for peptides obtained from in silico digestion of glutelin and prolamin of rice (*Oryza sativa*).

Fraction	Peptide	Molecular Weight (g/mol)	Ilogp	HBDs	HBAs
Glutelin	IVPQH	592.69	2.25	7	9
	PIVF	474.59	2.36	7	6
	IIQGR	585.70	1.28	10	9
Prolamin	QPY	406.43	1.34	5	7
	QSPVF	576.64	2.32	7	9
Control	GW9662	276.68	1.81	1	3

Data presented as the mean, n = 3. iLOGP (Octanol/water partition coefficient); HBDs (Number of hydrogen bond donors); HBAs (Number of hydrogen bond acceptors).

**Table 5 pharmaceuticals-16-00440-t005:** Protein fractions released after proteolysis.

Protein	Hydrolyzed Fractions
Glutelin	M-ASIN-R-PIVF-F-TVCL-F-L-L-CDGSL-AQQL-L-GQSTSQW-QSSR-R-GSPR-GCR-F-DR-L-QAF-EPIR-SVR-SQAGTTEF-F-DVSN-EL-F-QCTGVSVVR-R-VIEPR-GL-L-L-PH-Y-TN-GASL-VY-IIQGR-GITGPTF-PGCPETY-QQQF-QQSGQAQL-TESQSQSH-K-F-K-DEH-QK-IH-R-F-R-QGDVIAL-PAGVAH-W-CY-N-DGEVPVVAIY-VTDIN-N-GAN-QL-DPR-QR-DF-L-L-AGN-K-R-N-PQAY-R-R-EVEEW-SQN-IF-SGF-STEL-L-SEAF-GISN-QVAR-QL-QCQN-DQR-GEIVR-VER-GL-SL-L-QPY-ASL-QEQEQGQM-QSR-EH-Y-QEGGY-QQSQY-GSGCPN-GL-DETF-CTM-R-VR-QN-IDN-PN-R-ADTY-N-PR-AGR-VTN-L-N-SQN-F-PIL-N-L-VQM-SAVK-VN-L-Y-QN-AL-L-SPF-W-N-IN-AH-SIVY-ITQGR-AQVQVVN-N-N-GK-TVF-N-GEL-R-R-GQL-L-IVPQH-Y-VVVK-K-AQR-EGCAY-IAF-K-TN-PN-SM-VSH-IAGK-SSIF-R-AL-PTDVL-AN-AY-R-ISR-EEAQR-L-K-H-N-R-GDEF-GAF-TPL-QY-K-SY-QDVY-N-VAESS
Prolamin	M-K-IIF-F-F-AL-L-AEAACSASAQF-DAVTQVY-R-QY-QL-QQQM-L-SPCGEF-VR-QQCSTVATPF-F-QSPVF-QL-R-N-CQVM-QQQCCQQL-R-M-IAQQSH-CQAISSVQAIVQQL-QL-QQF-SGVY-F-DQAQAQAQAM-L-GL-N-L-PSICGIY-PSY-N-TVPEIPTVGGIW-Y

## Data Availability

Not applicable.

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
