# Peer review of "In Silico Identification of Peptides with PPARγ Antagonism in Protein Hydrolysate from Rice (Oryza sativa)"

_pharmaceuticals, 2023, doi:10.3390/ph16030440_

Round 1

Reviewer 1 Report

This manuscript describes the in-silico identification of ligands with PPARγ antagonism in protein 2 hydrolysates from rice (Oryza sativa). The article is overall well written and considered for publication, yet minor comments needed to be considered:

1.     In the title of manuscript is too long; in silico should be written in italic.

2.     In abstract (line # 20), you should mention PPARγ stands for what

3.     In the introduction part, the authors should also consider the recent published articles in the field of natural products and plant extracts in the management of different diseases such as:

-"Neuroprotective Effect of Artichoke-Based Nanoformulation in Sporadic Alzheimer’s Disease Mouse Model: Focus on Antioxidant, Anti-Inflammatory, and Amyloidogenic Pathways" Pharmaceuticals, 2022.

- "Chemical Composition, Antiaging Activities and Molecular Docking Studies of Essential Oils from Acca sellowiana (Feijoa). Chemistry & Biodiversity,2022.

4.     It is important to mention what are the future perspectives regarding this study.

Author Response

Dear reviewer, thank you for the comments. We appreciate every one of them.

About your valuable comments, the team answers.

  1. In the title of manuscript is too long; in silico should be written in italic. Thank you for the observation; we modified "in silico", and the title is shorter now.
  2. In abstract (line # 20), you should mention PPARγ stands for what. The abstract was remade due to your comments and the other reviewers' comments. Thank you for the observations.
  3. In the introduction part, the authors should also consider the recent published articles in the field of natural products and plant extracts in the management of different diseases such as:

-"Neuroprotective Effect of Artichoke-Based Nanoformulation in Sporadic Alzheimer’s Disease Mouse Model: Focus on Antioxidant, Anti-Inflammatory, and Amyloidogenic Pathways" Pharmaceuticals, 2022.

-"Chemical Composition, Antiaging Activities and Molecular Docking Studies of Essential Oils from Acca sellowiana (Feijoa). Chemistry & Biodiversity,2022.

Thank you for the suggestions about these scientific articles. Regarding the citation of published papers in the field of natural products, our manuscript referenced reports related to the characterization of bioactive peptides from the digestion of edible plants and their potential interaction with PPAR gamma. Thank you for the suggestion.

  1. It is important to mention what are the future perspectives regarding this study. Thank you for the observations. The team considers them very valuable. The conclusion and discussion were improved attending to this observation. 

Please see the attachment for the manuscript review.

Kindly regards.

Reviewer 2 Report

This manuscript describes the in-silico identification of ligands with PPAR gamma antagonists with different peptides with length from 2 to 5 amino acids under the in silico digestive process.

Page 7, For the docking setup, however, in Method section 4.5 and 4.6, it is unclear what the ionization states of the ligands are (short peptides).

Table 3 (page 4), the score bonding of short peptides are from -168 to -188 kcal/mol, however, the bonding energy, ranging from -1.8 to 6.3 kcal/mol, whereas the control GW9662 showed scoring bonding: -80.2 and bonding energy: -7.98 kcal/mol. Whats the difference between scoring bonding and bonding energy? Which one is used to measure the binding affinity between ligand and proteins? It is interesting to observe that the short peptides (Table 3) with 3-5 amino acids, having larger MW, but having less bonding energy than GW9662. Why the bonding energy between IVPQH and QSPVF shows such a large difference, -1.87 vs. -6.38 given that PQV are the same, the only difference is IH to SF. Can the authors provide an explanation why?

Author Response

Dear reviewer, thank you for the comments. The team considered every one of them. They help us for the improvement of the manuscript. 

This manuscript describes the in-silico identification of ligands with PPAR gamma antagonists with different peptides with length from 2 to 5 amino acids under the in silico digestive process.

Page 7, For the docking setup, however, in Method section 4.5 and 4.6, it is unclear what the ionization states of the ligands are (short peptides). Thank you for the observation. In this section, we improved the description of the methodology described in the 4.5 and 4.6 sections, trying to clarify this point. Between 807 to 810 lines, page 8, we describe these conditions.  

Table 3 (page 4), the score bonding of short peptides are from -168 to -188 kcal/mol, however, the bonding energy, ranging from -1.8 to 6.3 kcal/mol, whereas the control GW9662 showed scoring bonding: -80.2 and bonding energy: -7.98 kcal/mol. What’s the difference between scoring bonding and bonding energy? Which one is used to measure the binding affinity between ligand and proteins? It is interesting to observe that the short peptides (Table 3) with 3-5 amino acids, having larger MW, but having less bonding energy than GW9662. Why the bonding energy between IVPQH and QSPVF shows such a large difference, -1.87 vs. -6.38 given that PQV are the same, the only difference is IH to SF. Can the authors provide an explanation why?

Thank you for these significant comments. We reviewed the manuscript to improve and clarify the observations. Table 3 was modified, clarifying the GW9662 value, it was a mistake, and we appreciate your observation. The modification is on page 4, between 365 and 366 lines (Table 3).

Furthermore, the answer about scoring bonding and bond energy is now described between 347 - 361 lines, page 4, and discussed on page 5, 640 - 646 lines. Also, your interesting doubt about changes between IH to SF; in the PPARγ ligand binding pocket, a large hydrophobic region makes contact with the ligand, and Phenylalanine is shown as the most hydrophobic amino acid based on this approach. Page 6, 693 - 697 lines mentioned. 

Finally, the manuscript was improved (abstract, methodology, results, discussion, and conclusions) due to your valuable observations and the other comments of the reviewers. Please see the attachment.

Kindly regards.
